# RNA-Seq Analysis Reveals Expression Regulatory Divergence of W-Linked Genes between Two Contrasting Chicken Breeds

**DOI:** 10.3390/ani12091218

**Published:** 2022-05-09

**Authors:** Hongchang Gu, Liang Wang, Xueze Lv, Weifang Yang, Yu Chen, Kaiyang Li, Jianwei Zhang, Yaxiong Jia, Zhonghua Ning, Lujiang Qu

**Affiliations:** 1National Engineering Laboratory for Animal Breeding, Department of Animal Genetics and Breeding, College of Animal Science and Technology, China Agricultural University, Beijing 100193, China; guhc@cau.edu.cn (H.G.); ningzhh@cau.edu.cn (Z.N.); 2Beijing Municipal General Station of Animal Science, Beijing 100107, China; wangliangcau@139.com (L.W.); lvxueze0310@163.com (X.L.); carspstp@126.com (W.Y.); chenyu.cncn@163.com (Y.C.); likaiyanga@163.com (K.L.); zjw7432@126.com (J.Z.); 3Institute of Animal Science, Chinese Academy of Agricultural Sciences, Beijing 100193, China; yaxiongjia@163.com

**Keywords:** cis, trans, hybridization, regulatory evolution, w chromosome

## Abstract

**Simple Summary:**

Understanding the mode of gene expression and regulation is essential for understanding the evolutionary process. Many previous studies tried to explain regulatory changes at the autosomal level, but little research has extended these explorations to the field of sex chromosomes due to their complex sex-limit features. Here, we first adopted an innovative method of identifying regulatory divergence of W-linked genes. Compared with cis-regulatory divergence, trans acting genes were more extensive in the W chromosome. We also found that divergent sex specific selection cannot strongly affect the expression evolution of the W chromosome. This insensitivity to selection may be one of the reasons why regulatory divergence is so small between autosomal and sex chromosomes.

**Abstract:**

The regulation of gene expression is a complex process involving organism function and phenotypic diversity, and is caused by cis- and trans- regulation. While prior studies identified the regulatory pattern of the autosome rewiring in hybrids, the role of gene regulation in W sex chromosomes is not clear due to their degradation and sex-limit expression. Here, we developed reciprocal crosses of two chicken breeds, White Leghorn and Cornish Game, which exhibited broad differences in gender-related traits, and assessed the expression of the genes on the W chromosome to disentangle the contribution of cis- and trans-factors to expression divergence. We found that female-specific selection does not have a significant effect on W chromosome gene-expression patterns. For different tissues, there were most parental divergence expression genes in muscle, and also more heterosis compared with two other tissues. Notably, a broader pattern of trans regulation in the W chromosome was observed, which is consistent with autosomes. Taken together, this work describes the regulatory divergence of W-linked genes between two contrasting breeds and indicates sex chromosomes have a unique regulation and expression mechanism.

## 1. Introduction

Changes in genetic architecture can affect gene expression and their protein products, and further shape phenotypic variation [1,2,3]. As a form of genetic change, the variability of gene expression, referred to as transcriptional regulatory factors [4,5,6], were categorized as cis-regulatory elements and trans-regulatory factors. Cis-regulatory elements are present in the vicinity of genes on the same molecule of deoxyribonucleic acid (DNA), whereas trans-regulatory elements can regulate or modify genes distant from the gene from which they were transcribed by combining with their target sequences [7,8]. The persistence of cis- and trans-acting, and the mechanism by which they occur is particularly important because they play a large role in gene-expression novelty and phenotypic mutation [9,10]. The current approach that distinguishes regulatory changes in the animal genome is based on alleles expression divergence with fixed parameters, briefly, this method estimates differential expression both between the two parents and between the two alleles of the hybrid progenies [7]. To be specific, for autosomal in diploid individuals, the effect of cis-regulatory elements is allele-specific and quantified as the additive inheritance of genes [7,11,12]. By comparison, trans-regulatory factors regulate both alleles derived from parental breeds, and thus hybrid alleles cannot inherit expression level origin from parents respectively, so trans-regulatory divergence is enriched for a dominant effect [13]. However, due to the nature of sex-related inheritance of sex chromosomes, strategies need to be adjusted to identify the relative expression levels of “alleles” to assess regulatory changes.

Birds have a special sex chromosome system in which females have heterogametic sex chromosomes (ZW in females and ZZ in males) [14,15,16]. Bird W chromosomes lack recombination except for the pseudo-autosomal regions (PARs) [17], thus recombination suppression between the Z/W chromosome determines the female W-linked genes (except for genes located in PARs) existing independently and without alleles [18]. The analyzed methods that identify gene regulatory divergence and/or inheritance patterns in diploid can provide us with analogy insights for studying the regulatory changes of sex-limited W chromosomes [19,20,21,22,23]. Theoretically, since there is no concept of allelic expression, there may be discrepancies in the criterion that categorizes regulatory divergence in the W chromosome. Although the standard method to categorize regulatory variations could not apply in this study, the strategy of this method that identified differential expression between the two purebred progenies and hybrids could be referenced. Specifically, when hybrids’ expression level inherits from their mother, we think cis-regulatory elements act on W-linked genes, while the remaining W-linked genes were classified as subject to trans-regulatory effects. In other words, trans-regulatory divergence in the W chromosome always causes inconsistencies in expression levels between female progeny and her maternal parent. For regulatory changes, the inheritance pattern and the mechanism of regulatory divergence are often related. In the W chromosome, the effect of cis-regulatory variants is maternal dominance, whereas trans-regulatory differences are more likely to play a role in shaping all patterns except maternal dominance.

Modern chickens have been subjected to artificial directional selection [24,25,26], and thus they were exposed to different sex-specific effects, finally resulting in many phenotypic differences among breeds, such as egg number, body size, and female fecundity [27,28]. This divergence in reproductive capacity is also related to female fitness, and therefore affects W-linked genes expression and their inheritance. The artificial multi-events that have known the explicit direction led to rapid change under the domestication of the chicken also offered us an ideal model for revealing the relative contribution of the cis- and trans-regulatory variation in W-linked genes. Here, we used two chicken breeds, White Leghorn (WL) and Cornish (Cor), which have undergone varying sex-specific selection, to assess the parental expression difference and the role of two regulatory variations of W-linked genes in the brain, liver, and muscle at 1-day-old.

## 2. Materials and Methods

### 2.1. Samples

We used WL and Cor chickens from the National Engineering Laboratory for Animal Breeding of the China Agricultural University, as representative breeds of layers and broilers, respectively, to obtain pure-bred and hybrid progeny. Six offspring (three males and three females) were selected from the progeny of each mating, except for the Cor male × WL female cross, for which only two female offspring were obtained. In order to ensure consistency in the number of samples and the sampling time, thereby minimizing differences due to distant kinship between Cor males and females, we used two groups of three males and three females obtained from Cor male × Cor female and WL male × WL female, still designated as Cor and WL, respectively, to represent the parents of the hybrid offspring. We used artificial insemination to ensure consistent incubation times. Ultimately, we obtained 23 1-day-old chicks for tissue sampling (Figure 1).

Three tissues, including brain, liver, and breast muscle tissues were collected from all samples. We used CL to denote the F1 hybrid of Cor female and WL male, and LC to denote the F1 hybrid of WL female and Cor male (Figure 1). The chicks were euthanized with high-concentration carbon dioxide, which makes the animals lose consciousness quickly and minimize pain. As one-day-old chickens are too small for intravenous injection, we did not use pentobarbital sodium injection for euthanasia. Since we only focus on the regulatory divergences of the W chromosome, in the study section of the regulatory pattern, we only retain female individuals, and the expression of parental males is replaced by the expression of female individuals of the same breeds.

### 2.2. RNA-Seq and Principal Component Analysis

The tissues were deposited in RNAlater (Invitrogen, Carlsbad, CA, USA), an RNA stabilization solution, at 4 degrees Celsius for one night and then moved to −20 degrees Celsius refrigerator, and we extracted total RNA using Trizol reagent (Invitrogen, Carlsbad, CA, USA). The total RNA was sequenced on the Illumina HiSeq 2500 platform (Illumina Inc., San Diego, CA, USA) with 100-bp paired-end reads and 300-bp insert size. Finally, we obtained a total of 246.3 Gb of RNA-seq data, corresponding to an average of 3.6 million mappable reads per sample.

RNA-seq raw data were filtered using Fastp [29]. High-quality reads were aligned to the chicken reference genome (GCA_016699485.1) using Hisat2 [30,31]. After that, we used Stringtie [32] to estimate high-quality transcript abundance, with the normalization methods of Transcripts Per Kilobase Million (TPM) [33]. To investigate the patterns of gene expression for expressed W-linked genes in females, we analyzed the expression clusters of the three tissues by principal component analysis (PCA), as implemented in RStudio and visualized by R packages, factoextra, and FactMineR [34]. Before PCA, we removed genes with TPM < 0.5 in all samples to ensure the reliability of the results.

### 2.3. Classification of Cis- and Trans-Regulatory Categories

We used two inbred chicken breeds (Cor and WL) to generate reciprocal cross progeny. Since there is no W chromosome existing in the male genome, males are not considered in this analysis, therefore, we could classify by cross when identifying regulatory categories. Cor female, WL female, and CL female were put into one group (Group 1) and WL female, Cor female, and LC female were put into another group (Group 2). After determining the group, we removed all males for the further analysis of regulatory divergence.

Statistical thresholds of fold change >1.25 (or <0.8) and false discovery rate (FDR) < 5% were both set so that we could determine if there were significant differences in expression level [13,35]. The specific calculation method of fold change is the ratio of average TPM expression levels between breeds (for example, TPM_WL_/TPM_Cor_ > 1.25 or <0.8 means there was a significant difference between parents for transcriptome abundance). The false discovery rate was controlled by adopting a method of q-value estimation to correct the p-values of both the binomial test and Fisher’s exact test. The expressed genes (only considering females) were classified into three main categories according to the following criteria:(1)Cis: Significant difference between parents (WL and Cor), no significant difference between F1 and their maternal parents (CL and Cor; LC and WL), a significant difference between F1 and their paternal parents (CL and WL; LC and Cor). All “Cis” were considered to have “dominant” inheritance, i.e., expression of the F1 hybrid was only biased towards a single parent (Figure 2a,b).(2)Trans: Significant difference between parents (WL and Cor), a significant difference between F1 and their maternal parents (CL and Cor; LC and WL). The expression relationship between F1 and their paternal parent does not need to be considered. Considering the different inheritance patterns, “Trans” is also subdivided into “dominant” (Figure 2c,d), “additive” (Figure 2e,f), “over-dominant” (Figure 2g), and “under-dominant” (Figure 2h). Literally, “additive” indicates the genes for which expression in the hybrids was between the expression levels of the two parents. Genes for which the expression in hybrids was higher or lower than that in both parents were regarded as “over-dominant” and “under-dominant”.(3)Conserved: Significant difference between parents (WL and Cor), no significant difference between F1 and their maternal parents (CL and Cor; LC and WL), no significant difference between F1 and their paternal parents (CL and Cor; LC and WL).

Genes that were not significantly different between parents were not taken into consideration in this more detailed classification, because the effect of cis/trans would be masked due to their expression pattern. Regulatory divergences for the three tissues (brain, liver, and muscle) were identified separately, according to the threshold criteria described above.

## 3. Results

### 3.1. Divergence in Gene Expression among Parental Breeds

We chose two chicken breeds, Cor and WL, which exhibit varying female fecundity, responding to different sex-specific selection regimens, to represent the effects of elevated and reduced female-specific selection, respectively.

We characterized differential gene expression between the two parental breeds, and the number of deferential genes in the brain, liver, and muscle of the whole genome, were 1295, 3206, and 2623. We finally obtained 162 expressed W-linked genes in all three tissues, including 54 W-linked genes per tissue. There were 70 differential expressed genes (DEGs) (fold change > 1.25, FDR < 0.05), and the corresponding numbers in the brain, liver, and muscle were 17, 21, and 32 (Table 1, Appendix A). Most DEGs (87%) had a less than two-fold difference in expression, indicating that the expression divergence was subtle (Figure 3). Moreover, contrary to our conjecture, only 12 DEGs showed greater expression in WL (elevated female-specific selection breeds) than Cor (decreased female-specific selection breeds), including 1 uncharacterized gene and 11 protein-coding genes (*SKA1*, *SMAD7B*, *SPIN1L*, *MIER3*, *Myo5bl*, *SMAD4*, *MBD2*, *MdNAD-ME1*, *RL17*, *KCMF1L*, *TCF4L*). Notably, these 12 genes are the union rather than the intersection of WL-skewed expressed genes in the three tissues. We have investigated these WL-skewed DEGs based on GO analysis and existing research, the biological functions of them basically involve the activity and regulation of DNA, RNA, and protein, and no genes are associated with reproductive traits. There was a stark contrast in the expression of the W-linked genes between three tissues, convergent patterns of gene expression were detected in the brain, only 31% of the genes in the brain showed significant divergence, while genes in muscle showed more discrete expression, of which 59% of them were classified as DEGs.

### 3.2. Different Expression Clustering Patterns across Tissues and Populations

For each hybrid cross, we collected RNA-seq data from the brain, liver, and muscle tissue of 23 F1 progenies 1-day post-hatching. On average, we recovered 29.17 million mappable reads per sample, genome-wide differential expression profiles are first identified to identify potential signatures of the data. We only kept female individuals for PCA analysis. According to the criteria in the previous step, the unexpressed genes had been removed. We observed significant differences in the transcriptome profile between different tissues and between maternal-origin by PCA results (PERMANOVA and pairwise comparison, *p*-adj < 0.05). Tissue was the most significant factor acting on W-linked gene expression (Figure 4A), genes from different tissues were clustered separately and fell within different circles. For the brain and liver, the maternal origin of Cor/CL rather than WL/LC seemed the most powerful because samples were clustered based on it (Figure 4B–D). While for the muscle tissue, samples were aggregated according to breeds. These patterns of separation and aggregation were basically in line with our expectations.

### 3.3. Contribution of Cis- and Trans-Acting Effects Based on the Classification of Regulatory Divergence

Both the parental and hybrid data sets were analyzed for evidence of differential expression using the binomial exact test and fold change parameters. According to the differential expressed situation, expressed genes were classified into different categories of regulatory divergence (Figure 5, Table 1, Appendix A). In our experimental conditions, trans-acting genes were more extensive compared with cis-regulatory divergence, with averages of 68.6% vs. 31.4% in all three tissues. Specifically, there are 17 DEGs in the brain, 11 (64.7%) of which are classified as “Cis” in Group 1, and none of them are classified as “Cis” in Group 2 (Table 1). This apparent group-specific pattern was not present in the liver and muscle. In the liver, there were 38.1% “Cis” in group 1 and 33.3% “Cis” in group 2, which were 34.4% and 21.9% in muscle, respectively (Table 1). More than half of the genes exhibited dominant expression (brain: 61.8%; liver: 66.7%; muscle: 53.1%), and we speculate that it might be due to the strong effect of the specific allele from Cor or WL. To verify our conjecture, we further examined the W-linked genes for evidence of skewed expression of one allele. All 21 expressed dominant genes showed Cor-skewed dominance in the brain. Such biased inheritance patterns are not extreme in muscle (around 62%). Interestingly, this strong heredity power of the Cor allele seems to lose its efficacy in the liver and around 52% of dominant genes showed Cor-skewed. Heterosis widely exists in hybridization events, corresponding to the improvement of the production performance of hybrids [36,37]. Although hybridization disadvantages occasionally appear, they are avoided in the breeding process as much as possible. As expected, we observed heterosis was ~three times more common than the hybrid disadvantage. Specifically, up-regulation of W-linked genes in hybrids was far more common in the muscle, 14 expressed genes showed over-dominance. Only one and four genes were expressed at higher levels compared with both parents in the brain and liver. Within tissues, few of these W-linked genes showed consistent regulatory divergence category between two groups, the proportion of these genes accounted for around 18%, 10%, and 25% of the total in the brain, liver, and muscle.

## 4. Discussion

Previous studies found that the direction and magnitude of sexual selection can partially shape the evolution of gene expression on the W chromosome [27]. Two breeds were selected as the samples representing distinct sexual selection modes. Given its differences in reproductive performance and other female fitness traits, we expected apparent divergences of W-linked transcriptome abundance also existed between WL and Cor. Surprisingly, our result was the opposite of the assumption in all three tissues. Specifically, only less than half of the expressed genes (43%) showed significantly different expression in the three tissues, and 17% of these genes are WL-skewed. For 12 WL-skewed genes, we had not identified biological functions associated with female fecundity. Interestingly, these genes were enriched in pathways related to cell division and metabolism. We inferred that there are three main reasons: First, biased expression is not necessarily a fixed property of genes, expression levels can vary greatly among tissues [38,39,40,41]. Generally, somatic tissues show much less dimorphism than gonads [42,43,44]. We chose three somatic tissues instead of gonads as the experimental samples because somatic cells are subjected to a purer selection force, and the “net effect” of gene expression is more meaningful. Second, gene expression is also highly variable over the course of development. Previous studies showed there are expression changes with minor divergence in embryonic stages and high-level divergence in sexually mature adults [45,46]. Nonetheless, we still selected 1-day-old chicks because a previous study has demonstrated that female-specific selection in birds is strongest during this developmental time point [45]. Third, known W-linked genes do play an important role in sex determination, but there is no evidence that their functions are patently associated with sexual fitness [47], the shaping effect of sexual selection on the W chromosome may also be insignificant.

Although the assumption of parental expression divergence deviated from our original intention, it did not affect the procedure and results of our exploration of regulatory changes. Before identifying regulatory variations, we observed expression clusters between tissues, and between breeds [48,49]. The different sample clusters by tissue indicated that tissues played the most significant role in gene expression for all individuals. Within tissues, the ancestry-specific patterns could both be observed clearly and typically in muscle. We speculated that this obvious pattern in muscle might be related to the disparity in growth and development performance between breeds. Our results suggested that maternal origin was an important contributor to the cluster in gene expression in our dataset. This clustering feature was in line with our prediction due to the maternally inherited mode of the W chromosome. Our results also indicate that this maternal origin mode is more obvious in Cor/CL, but its shaping effect in WL/LC did not seem to be significant. This evidence proved that Cor W-linked genes had stronger hereditary capabilities from another perspective. The PCA overview results proved the genetic differences between the reciprocal crosses, and further demonstrated the necessity of identifying the regulatory divergence according to it.

Both regulatory changes and inheritance patterns are based on gene expression dynamic changes in the hybridization process [13,50]. Mechanisms of regulatory divergence may influence the inheritance of gene expression, and recent studies showed that inheritance diversity may depend on the effects of trans-regulatory factors of one genome on the other genome [11,51,52], and hence the regulatory divergence between parental species. This assumption is based on the effect that dominance is caused by trans-regulatory factors in diploid hybrids. Genes located in the non-recombination region of the W chromosome (NRW) can be regarded as a ‘single allele’, so cis-/trans- regulatory elements can also lead to a dominant pattern. The difference is that cis-regulatory divergence will lead to a maternal-dominance, while trans-regulatory divergence contributes independently to paternal-dominance. Nevertheless, a large proportion of expressed genes showed dominance (both including dominant genes caused by cis-/trans-regulatory effects), and previously plant-based studies showed that the dominant pattern is prevalent and widespread among different natural populations, and maybe also closely related to the phenotypic novelty of hybrids [53,54,55]. The above evidence proved two conclusions: First, the special sex-limit characteristics of the W chromosome will cause a different regulatory changes mechanism compared with autosomes and Y chromosomes. Second, the result that the novel or the superiority phenotype has a certain link to dominant patterns in offspring is consistent with the classic concept of hybrid breeding.

Most classified genes were controlled by trans-regulatory factors rather than cis-regulatory elements, the ratios of “Trans” are 0.59, 0.57, and 0.68 in the brain, liver, and muscle, respectively [11]. These results were consistent with the previous observation in the autosomal genome. These observations confirmed that the gene expression evolution of most W-linked genes might be controlled by loci on autosomes or Y chromosomes. The similar effect of regulatory changes on autosomal genes and sex-linked genes showed that the total efficacy of regulatory divergence along the entire genome was basically stable. To identify the relative contribution of the “W single allele” originating from WL and Cor, we carefully observed whether there was a breed-skewed expression mode in dominant W-linked genes. Interestingly, all dominant genes in brain tissue exhibited Cor-skewed, regardless of the female paternal expression level. This extreme imbalance was not observed in liver and muscle tissues, the proportion of Cor-skewed expression genes only accounted for 52% and 62% respectively, and only showed a slight advantage over WL-skewed expression. We previously thought that hybrids might inherit the excellent muscle growth characteristics of the Cor parent and thus, be more Cor-skewed compared with the other two tissues. For this result that was contrary to our conjecture, there were at least three, non-mutually exclusive, possible explanations. First, compared with the liver and muscle, the brain has the most conservative expression pattern [56], the expression regulation was not easily affected by reciprocal crosses and has a high consistency. Second, heterosis in hybrids may be mainly reflected in over-dominance rather than dominance [56,57]. The evidence for this inference in this study was that muscles have the most “over-dominant” genes. Finally, unlike autosomal genes, the function of W-linked genes may be less related to muscle and body development.

Taken together, this study significantly improves our understanding of regulatory evolution on a sex-limit genomic scale. We drew on the traditional methods that distinguish between cis- and trans-acting sources on autosomes and used a new method to evaluate the regulatory changes of the W chromosome for the first time. Our results also provided a systematic look at the evolution of cis- and trans-acting, and incorporated the inheritance pattern into the same research framework with regulatory divergence. In principle, this joint analysis approach had advantages because of the causal relationship between these two concepts. Cis-regulatory elements and trans-acting factors control nearby and distant gene expression. Meanwhile, the hereditary architecture of gene expression levels determines the inheritance pattern. Using the RNA-seq data of the hybridization model, we globally identified the features at the transcriptome level of W-linked genes and visualized them through the classification of regulatory divergence. More instances of trans-regulatory divergence than instances of cis-regulatory divergence were observed in the W chromosome and this might be because of the relatively short divergence history of Cor and WL [25,58]. This low genetic diversity was also a potential cause that DEGs among parents only account for a small part of expressed genes.

What is certain is that although the regulatory pattern identifies based on the transcriptome level is reasonable, it does not mean that the result is completely accurate. A small number of organized, single-point, fixed-parent studies are not enough to allow us to understand the mechanism and laws of regulatory divergence from a broader perspective, also does not allow us to locate the sequence sites of these regulatory factors. What can be encountered is that when these limitations are broken, higher throughput, and more accurate sequencing methods are applied, the problem will be solved, and the research on regulatory divergence will not just stop at the stage of description and statistical analysis.

In conclusion, our research used innovative methods to identify the genetic pattern and regulatory divergence of the W chromosome, which was not limited to a single tissue, and a single set of conditions. The results revealed an autosomal-like regulatory model, which implied a robust mechanism of regulatory divergence across whole sequences. Insights on W-linked gene expression regulation and evolution would expand such research at the species (or breed) and genome levels.

## Figures and Tables

**Figure 1 animals-12-01218-f001:**
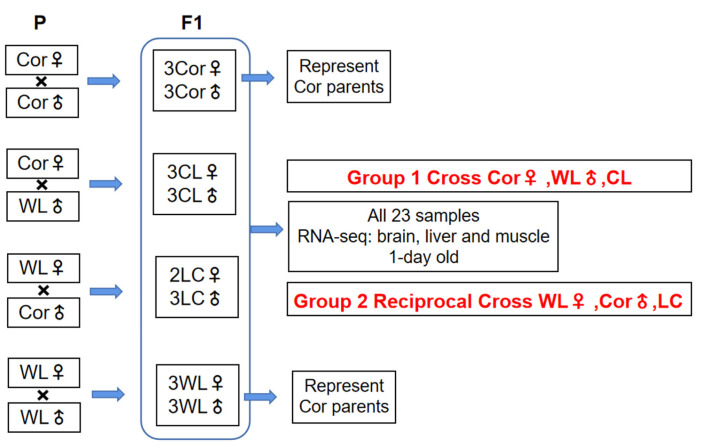
**Samples and experiment design.** Group 1 and Group 2 represent the cross and reciprocal cross, respectively. All analyses use female individuals for two groups in order to identify the expression of the W chromosome. Samples of three tissues (brain, liver, and muscle) were collected from all chicks one day after hatching for transcriptome sequencing.

**Figure 2 animals-12-01218-f002:**
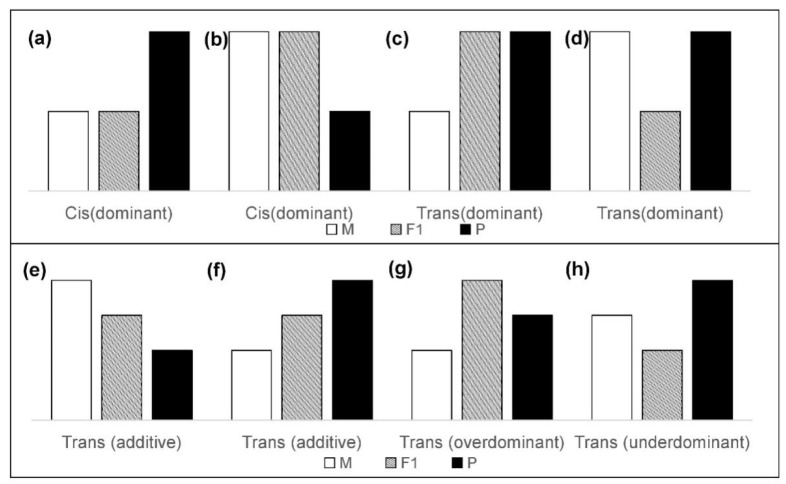
**Hypothetical classification of regulatory divergence.** Classification according to the expression level in the maternal parent (mother of CL: Cor/mother of LC: WL), paternal parent (father of CL: WL/father of LC: Cor), and F1 hybrids (CL/LC). Subdivide the ‘Cis’ (**a**,**b**) and ‘Trans’ (**c**–**h**) main categories are listed. “M”, “F1”, and “P” are the abbreviations for “mother”, “F1 hybrid”, and “female of father breeds” respectively.

**Figure 3 animals-12-01218-f003:**
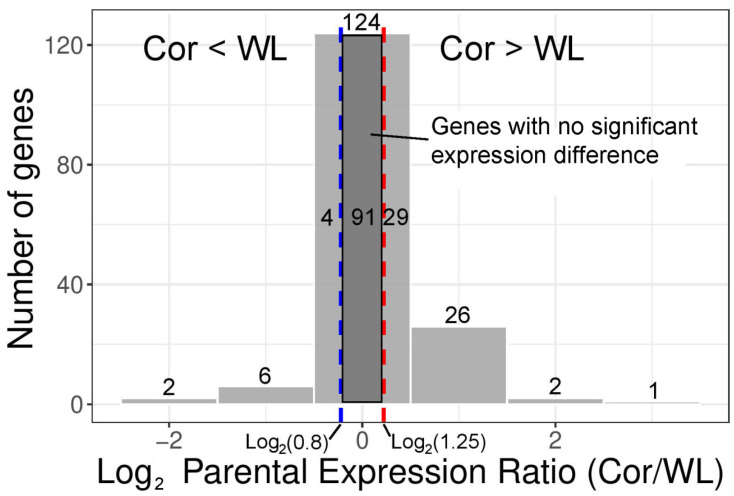
**Differences in gene expression between Cor and WL.** The histogram shows the direction and magnitude of changes in the expression in genes exhibiting divergent parental expression. The two vertical dashed lines represent the expression thresholds of parental divergence. Negative values indicate up-regulated expression of WL, and positive values indicate up-regulated expression of Cor.

**Figure 4 animals-12-01218-f004:**
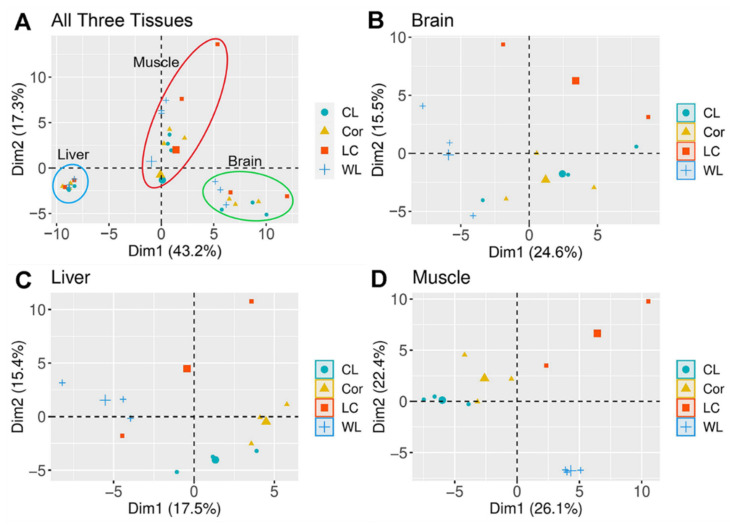
**Principal Component Analysis using RNA-seq data.** (**A**) PCA results of all three tissues, the sample is divided into 3 different clusters according to the 99% confidence interval. (**B**) PCA results of the brain. (**C**) PCA results of the liver. (**D**) PCA results of muscle. Each dot represents an individual, and dots with different shapes and colors indicate different varieties.

**Figure 5 animals-12-01218-f005:**
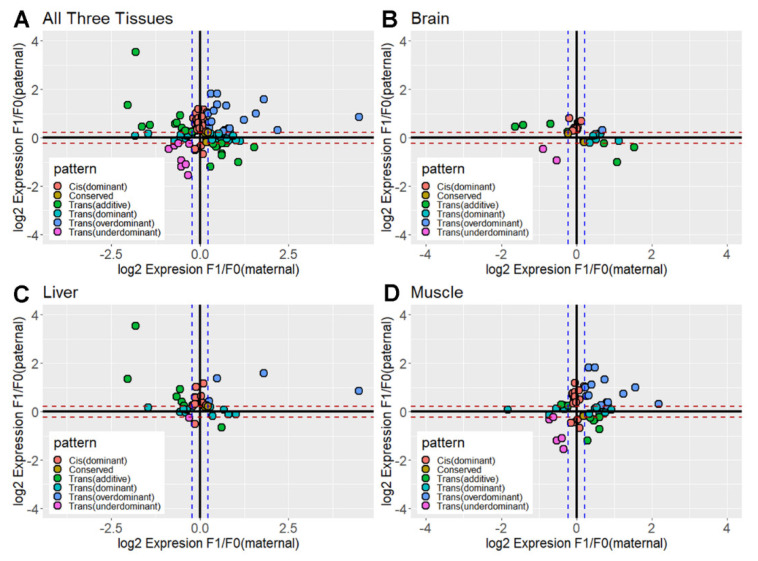
**The scatterplot compares the W-linked expressional differences between hybrids and their parental breeds.** The visualized results are respectively displayed in (**A**) all three tissues; (**B**) brain; (**C**) liver; and (**D**) muscle. The different colored dots represent the different regulatory divergence categories. Log_2_ Expression F1/F0(maternal) on the X-axis specifically includes two cases, namely log_2_ Expression CL/Cor or log_2_ Expression LC/WL, corresponding to the two cases on the Y-axis, log_2_ Expression CL/WL or log_2_ Expression LC/Cor.

**Table 1 animals-12-01218-t001:** The number of regulatory divergence categories of W-linked genes in three tissues.

Tissues	Groups	Number of Genes					
		Cis	Trans				Conserved
		Cis (Dominant)	Trans (Dominant)	Trans (Additive)	Trans (Overdominant)	Trans (Underdominant)	
Brain	1	11	0	4	0	1	1
	2	0	10	3	1	1	2
Liver	1	8	6	6	1	0	0
	2	7	6	1	3	1	3
Muscle	1	11	6	2	5	8	0
	2	7	10	7	4	2	2

## Data Availability

All raw data during the current study are available in the NCBI BioProject (https://submit.ncbi.nlm.nih.gov/subs/bioproject) with accession number PRJNA591354 (accessed on 24 November 2019).

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
