# Peer review of "RNA-Seq Analysis Reveals Expression Regulatory Divergence of W-Linked Genes between Two Contrasting Chicken Breeds"

_animals, 2022, doi:10.3390/ani12091218_

Round 1
Reviewer 1 Report
The proposed topic and the approach of analysis is interesting and innovative but the work is not clearly presented, the method of analysis and results are not adequately described mad incomplete, Many concepts are often redundant.
In the introduction the approach used and the objectives of the work must be defined better : in line 17-51 citation 11 does not appear correct and the argument must be explained better.
Materials and methods:
Paragraph 2.1 is unclear: explain the mating plan better, how many animals for parental lines? How they mated? How many chicks hatched, how many males and females. Do not anticipate the collection of tissue samples (line to 85), the groups of animals must be better defined. The sequencing method (line 96-98) would go in paragraph 2.2.
Paragraph 2.2 : line 104-108 is not clear, why to analyze W-linked genes it is necessary to analyze gene clusters with PCA
Paragraph 2.3 the experimental approach is unclear: what groups of animals are compared and what information is obtained. If males are not analysed this should already be defined during sampling (paragraph 2.1). Line 118-122 should be moved to the introduction . it should be explained better the strategy used for the cis-trans classification used in the work. Define how the fold change was calculated and between which groups of animals. Linea 124: what does expressed gene mean? All genes expressed or only those differentially expressed? Between which groups?
What does "CIS DOMINANT" mean, if you only analyze W-linked genes you have no alleles... explains better, maybe the difference of expression observed between the two breeds (parental line)? In this case the different expression is explained only by a cis regulation and the trans effect is absent? Line 132-133 because paternal parent should not be considered ? Explain, In Figure 2 it seems to be take in account.
Line 134-139 The classification of trans effects is not clear, which thresholds have been used, and which approach of data analysis? Line 140-145 “Conserved” is equal to Cis description (line 126-130)
Results
They are incomplete and unclear: line 148-153 is not a result, line 156-159: the results should also be reported separately for the 3 tissues as well as Figure 3. Figure 3 is not clear: it reports a different number of DEG than the text. What “no divergent expression genes” mean? and the LOG2 thresholds 0.8-1.25?
line 159 Which are the 12 genes? In which tissues? What Fold changes?
3.2 is not clear. Part of the information goes in paragraph 3.1 and part in the discussion.
Line 172 the differences between tissues are not reported. The differences are referred to set of genes expressed or different expression values between the two parental lines? It is not clear. The results are not clearly expressed, how much and which genes are different?
3.3 The results of the cis-trans effect analysis shall be reported for the individual genes analyzed and separately for each tissues. The text is largely a discussion, the data analysis is not reported.
Author Response
Response:
Many thanks to the reviewers for your questions and comments on sampling, analysis and results, which helped a lot in the improvement of our manuscript, and we carefully backtracked our research and answered and resolved these questions.
introduction
- In the introduction, we have corrected some of the description, and updated and added citations to the bibliography on lines 47-51 to better explain the arguments. In addition, we also rewrite some definitions that are not clear, and delete concepts and statements that may be repeated.
Material and methods
- In terms of sample collection, we have redrawn the figure (figure 1) about the experimental design, which improved the experimental design scheme and the number of samples. At the same time, we have also revised this part of the content in the text. The specific mating plan, and the number of samples of parents and offspring are all explained in detail. Additionally, we moved the initial part of RNA-seq to paragraph2-2.
- Before performing sex chromosome regulatory pattern analysis, we analyzed the whole transcriptome data feature. There are two main purposes for this analysis.
First, we performed PCA analysis for the 3 tissues separately, and also performed this analysis for all samples, the tissue specificity of the clustering pattern proves that we are considering the case of tissue differences It makes sense to analyze regulatory changes below.
Second, the CL and LC populations are not clustered together in the PCA graph, which could also be a potential explanation for why regulatory divergences have mating pattern (cross or reciprocal cross) specificity.
- First, we adjusted the structure of 2-3 as you suggested, moved the description of using only female individuals in the analysis of the regulatory changes to sections 2-1, moved the description of the specificity of the W chromosome to the introduction section and rewritten it to read into the whole paragraph.
We have rewritten the definition of cis/trans analysis, i.e. fold change, significance test, and tissues. Expressed genes are previously defined genes with a TPM greater than 0.5 in all tissues (2-2 line123-124: "we removed genes with TPM < 0.5 in all samples to ensure the reliability of the results."). This expression can be ambiguous, so we dropped "expressed"
- First, as you said, the W chromosome does not have Z-linked alleles, so we, therefore, redefine the previous method for identifying cis/trans in this study. Similarly, for autosomes, genes were classified as Cor dominant or WL dominant when the expression level in hybrids was only similar to that of the corresponding parent. Obviously, when analyzing the W chromosome, the expression level of the male parent variety is impossible to know. So here we also redefine, we take the average expression level of female individuals in the male parent breed as the male parent expression level, in order to identify dominant, additive and other inheritance patterns.This definition idea is exactly the same as the identification method of regulatory changes.
Line 134-139: We rewrote the description of significant difference judgments. The threshold parameters are described in more detail in this paragraph.
“Statistical thresholds of fold change>1.25 (or < 0.8) and false discovery rate (FDR) < 5% were both set so that we could determine if there were significant differences in expression level. The specific calculation method of fold change is the ratio of average TPM expression levels between breeds (for example TPMWL/TPMCor > 1.25 or < 0.8 means there was a significant difference between parents for transcriptome abundance). The false discovery rate was controlled by adopting a method of q-value estimation to correct the p-values of both the binomial test and Fisher’s exact test.”
The "Conserved" mode and "Cis" are actually different in definition. "Cis" defines a significant difference between F1 and their paternal parents ("paternal parents" here refers to females individuals of the paternal breed). While "Conserved" means that the gene expression of the F1 does not differ from that of both parents (here the expression of the paternal parent is still the same as above)
Results
- line 148-153 we have rewritten the sentence and moved it to the discussion section
Line156-159 To describe the detailed results in the three tissues, we supplement the results in this section as follows:
“We characterized differential gene expression between the two parental breeds, the number of deferential genes in the brain, liver, and muscle of the whole genome, were 1295, 3206, and 2623. We finally obtained 162 expressed W-linked genes in all three tissues, including 54 W-linked genes per tissue. There were 71 differential expressed genes (DEGs) (fold change > 1.25 , FDR<0.05), the corresponding numbers in the brain, liver and muscle were 16, 23 and 32 (Table 1).”
For figure 3, “no divergent expression genes” is an inappropriate description, We modified it to "genes with no significant difference" and marked it on the figure.
Because we previously defined the expression threshold of DEGs as fold change > 1.25 or < 0.8, the abscissa in figure.3 represents the log2Expression(Cor/WL), so the x-axis scale corresponding to the DEGs on the figure should be in the left of log2 (0.8) or the right of log2(1.25).
- We clearly describe the 12 genes, how they are defined, and their corresponding biological functions in the revised manuscript. Previous studies suggested that breeds with high female-specific selection may correspond to the high expression of W-linked genes. But our research doesn't seem to support this conjecture. We focused on these 12 W-linked DEGs and demonstrated by GO analysis that even WL-skewed expression genes do not correspond to female reproductive traits, and thus they are insensitive to the female-specific selection, which is consistent with our previous results.
This is our revised sentence:
“We finally obtained 162 expressed W-linked genes in all three tissues, including 54 W-linked genes per tissue. Also contrary to our conjecture, only 12 DEGs showed greater expression in WL (elevated female-specific selection breeds) than Cor (decreased female-specific selection breeds), including 1 uncharacterized gene and 11 protein-coding genes (SKA1, SMAD7B, SPIN1L, MIER3, Myo5bl, SMAD4, MBD2, MdNAD-ME1, RL17, KCMF1L, TCF4L). Notably, these 12 genes are the union rather than the intersection of WL-skewed expressed genes in the three tissues. We have investigated these WL-skewed DEGs based on GO analysis and existing research, the biological functions of them basically involve the activity and regulation of DNA, RNA and protein, and no genes are associated with reproductive traits.”
- We have reintegrated the results related to 3-2 to make it more complete and clear.
- Line172
I apologize for the mistake caused by the unclear sentence, the expression differences here are actually an intuitive reflection of the PCA results, i.e. the PCA plot (Figure 4A) shows a complete separation of tissue clusters. We reworded this sentence as “We observed significant differences in transcriptome profile between different tissues, between maternal-origin by PCA results.”
- We have considered your doubts, but except for the "single allele", the characteristic that the W chromosome distinguishes autosomes is the difference in gene counts. Autosomes have tens of thousands of genes, and there are less than 70 W-linked genes. When we identified DEGs and classified these W-linked DEGs so finely, the number of distinct regulatory divergences in a single tissue is not sufficient to allow us to identify its overall characteristics and even mislead our conclusions.At the same time, according to your suggestion, we will show the results of the inter-organization more concretely to make the results more substantial.
Reviewer 2 Report
The manuscript "RNA-seq analysis reveals expression regulatory divergence of W-linked genes between two contrasting chicken breeds" described the W-linked genes expression in brain, liver and breast muscle at 1-day-old chicks. Although the results were not expected from the authors' hypothesis, the current study presents a broader pattern of trans-regulation in the W-chromosome, which is an autosomal-like regulatory model.
The manuscript is in a good shape with reasonable levels of English. The limitations of the study with the restricted parents and single point samples may give us doubt of the study confidence. However, the authors are well aware of the above limitations and clearly explained them. The only minor thing the review prefer to see is biological functions associated with the W-linked DEGs in three tissues, even though the authors stated not to identified biological function.
Author Response
Thank you very much for your affirmation of this manuscript about its presentation of methods, results, and discussion. And we also gratefully appreciate for your valuable suggestion and comment. Thank you very much for your suggestion to modify the biological function description of DEGs, which we have explained in detail in the main text.
Reviewer 3 Report
An impressive study on regulatory divergence of W-linked genes between two contrasting chicken breeds. I have made some suggestions on the edited draft to guide the authors to improve on the quality of the paper.

Author Response
Thank you very much for your thorough review and evaluation of the manuscript.
We have checked and revised according to your markings in the manuscript, and I believe your suggestions will significantly improve the quality of the manuscript. At the same time, we also checked the article from beginning to end for possible grammar, description not clear and tense errors, and made comprehensive revisions. Thanks again for your detailed review!
Round 2
Reviewer 1 Report
The work was completed as required and in this form is much clearer and easier to read. There are still some points that can be improved:
LINE 108: Missing end of sentence
line 173: there is an inconsistency in table 1 respect to the text where 16 genes are reported for the brain and in table for group 2 the sum is17 genes.
paragraph 3.2 you reported differences among groups in PCA. it would be necessary to add a statistical test to verify that the differences are significant.
Table 1 : I propose to modify it to complete the exposure of the results: in the table for the 71 genes identified in 3 tissues the name of the gene, tissue, Fold change between parental lines, Regulatory category must be reported. (See an example in PDf attached)
Figure 5 caption: line 257 you white “WL female on the X-axis and COR female on the y-axis” But is it F1/F0 (maternal ) for X axis, as shown in the figure, that mens CL/Cor and LC/WL and F1/F0 (paternal) for y axis that is CL/WL and LC/Cor ?

Author Response
Thank you very much for your re-examination and suggestions for the article. We revise and supplement the article according to your review. Below are the answers and modifications for each question.
1 line 108, we added the end of the sentence, this is an error caused by proofreading negligence, so we re-read the full text to avoid this error.
2 We are very sorry for the wrong count due to our ignoring there is 1 "conserved" gene in group 1 of brain tissue. As you said, there are a total of 17 DEGs in the brain. At the same time, we checked the content of the full text involving the ratio and the gene counts to ensure that the calculation is correct.
3 Thank you very much for your suggestion, we calculated the difference between different groups of PCA as you requested, using PERMANOVA (Permutational multivariate analysis of variance) and pairwise comparison, p-adj<0.05 was considered as a significant difference between groups.
4 We aggregate information such as gene name, tissue, Fold change between parental lines, Regulatory category, etc. in one table. Thank you very much for attaching 1 PDF to illustrate your suggestion, again our results will be put into a table similar to this PDF. Since the table is larger, we put it in the attachment, i.e. Table S1.
5 As you said, we found an ambiguity in the legends of figure 5. So we revised the description in this section. The following is the modified content
“Figure 5. The scatterplot compares the W-linked expressional differences between hybrids and their parental breeds. The visualized results are respectively displayed in A) all three tissues; B) brain; C) liver; D) muscle. The different colored dots represent the different regulatory divergence categories. Log2 Expression F1/F0(maternal) on the X-axis specifically includes two cases, namely log2 Expression CL/Cor or log2 Expression LC/WL, corresponding to the two cases on the Y-axis, log2 Expression CL/WL or log2 Expression LC/Cor.”